# ROBO-INSTRUCT: Simulator-Augmented Instruction Alignment For Finetuning CodeLLMs

## Abstract

Large language models (LLMs) have shown great promise at generating robot programs from natural language given domain-specific robot application programming interfaces (APIs). However, the performance gap between proprietary LLMs and smaller open-weight LLMs remains wide. This raises a question: Can we fine-tune smaller open-weight LLMs for generating domain-specific robot programs to close the performance gap with proprietary LLMs? While SELF-INSTRUCT is a promising solution by generating a diverse set of training data, it cannot verify the *correctness* of these programs. In contrast, a robot simulator with a well-defined world can identify execution errors but limits the diversity of programs that it can verify. In this work, we introduce ROBO-INSTRUCT, which brings the best of both worlds — it promotes the diversity of SELF-INSTRUCT, while providing correctness of simulator-based checking. ROBO-INSTRUCT introduces ROBOSIM to synthesize a *consistent* world state *on the fly* by inferring properties relevant to the program being checked, and simulating actions accordingly. Furthermore, the instructions and programs generated by SELF-INSTRUCT may be subtly inconsistent — such as the program missing a step implied by the instruction. ROBO-INSTRUCT further addresses this with INSTALIGN, an instruction-program alignment procedure that revises the task instruction to reflect actual results of the generated program. Given a few seed task descriptions and the robot APIs, ROBO-INSTRUCT is capable of generating a training dataset using only a small open-weight model. This dataset is then be used to fine-tune small open-weight language models, enabling them to even exceed the performance of several proprietary LLMs including GPT-3.5-Turbo and Gemini-Pro.

## 1 Introduction

Large language models (LLMs) have demonstrated great promise at generating robot programs from natural language instructions [3, 10–12, 17, 18, 31, 39]. For example, consider an instruction for a service mobile robot: *"Check how many conference rooms have no markers."* The robot may be equipped with a domain-specific robot application programming interface (API) that includes skills such as `go_to(location)` for navigation and `is_in_room(object)` for perception. Since such domain-specific APIs do not exist in the training dataset of general-purpose LLMs, in-context learning (ICL) via few-shot examples is often employed to describe and use such APIs for performing few-shot inference. However, there is a significant performance gap [10] in the correctness of programs generated by ICL for large proprietary models and smaller open-weight models that can be deployed locally on robots. This raises a question: can we fine-tune *small open-weight LLMs* for generating domain-specific robot programs to close the performance gap with proprietary LLMs?

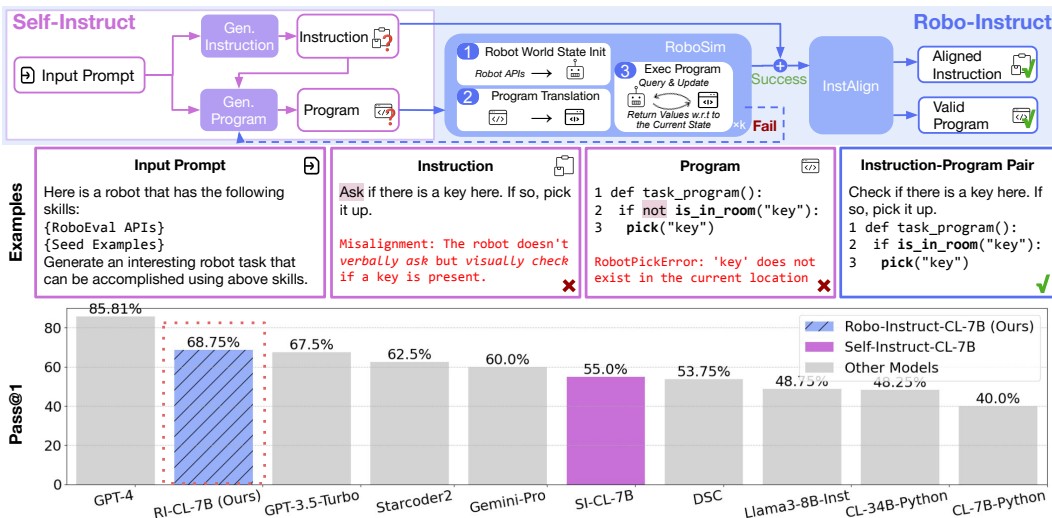

Figure 1: High-Level Overview of ROBO-INSTRUCT. This figure also illustrates an example of an invalid SELF-INSTRUCT-generated instruction and program, as well as pass@1 results of different LLMs on ROBOEVAL.

Since training datasets of the domain-specific robot programs are often unavailable, SELF-INSTRUCT might seem like a promising solution [29, 36]. Consider the setting of generating programs for service mobile robots that can perceive objects, navigate to various locations, manipulate items, and communicate with humans. By formulating these robot skills into APIs, we can create a few seed task examples demonstrating their use case and employ SELF-INSTRUCT to generate a diverse set of instruction-program pairs as training data, as illustrated in Fig. 1. However, using SELF-INSTRUCT naïvely may generate infeasible instructions—e.g., asking the robot to pick up multiple objects at once when it cannot due to physical constraints. They can also violate domain-specific constraints. For example, in Fig. 1, after line 2 confirms the absence of a key at the current location, line 3 erroneously attempts to pick up a key. Further, these instructions may not align with the generated programs, even if these programs are valid. For example, Fig. 1 shows an example instruction directing the robot to *verbally ask* in each room if a key exists, whereas the program instructs the robot to *visually check* in each room. Finally, the generated programs may have execution errors. These challenges may appear to be solvable using a simulator, but a simulator needs an initial world state to check against programs. A simulator using a hand-curated world state will end up rejecting the wide diversity of programs generated by SELF-INSTRUCT, even if they are executable, just because the world state did not capture some aspect relevant to them (*e.g.,* the presence of a "key").

This work introduces ROBO-INSTRUCT, a new framework based on SELF-INSTRUCT, to address these issues and improve the performance of small open-weight language models for generating domain-specific robot programs. As shown in Fig. 1, ROBO-INSTRUCT introduces two novel components: **(1)** ROBOSIM, a task-agnostic simulator that encodes domain-specific constraints and validates robot programs generated from SELF-INSTRUCT. Critically, ROBOSIM *dynamically* synthesizes a *consistent* world state starting from arbitrary programs. **(2)** INSTALIGN, an instruction-program alignment procedure that revises the generated instructions to better reflect the intent of the generated programs. ROBO-INSTRUCT also employs a rejection-sampling mechanism that rejects invalid programs detected by ROBOSIM and queries SELF-INSTRUCT for a new program corresponding to the same generated instruction.

We validate ROBO-INSTRUCT by fine-tuning Codellama-Python-7B [30] and evaluate on ROBOEVAL, a domain-specific code generation benchmark for service mobile robots. We show that ROBO-INSTRUCT is capable of improving the performance of the Codellama model by using only a small open-weight model to generate the training dataset. Compared to the base Codellama-Python-7B model without fine-tuning, our ROBO-INSTRUCT fine-tuned models outperform by $28.75\%$ in average pass@1 scores; and, compared to SELF-INSTRUCT fine-tuned model, our model outperform by $13.75\%$.; and the best pass@1 of ROBO-INSTRUCT fine-tuned model achieves a $68.75\%$ match, surpassing the performance of the proprietary GPT-3.5-Turbo and Gemini-1.0-Pro.

**Contributions** Our main contributions are as follows:

1. We introduce ROBO-INSTRUCT, a new framework for improving the code generation performance of small open-weight language models for domain-specific robot programs. This framework introduces two novel components, ROBOSIM and INSTALIGN.

2. We introduce a *dynamic world synthesis and evaluation* process for generating relevant world states for automated code checking for diverse, arbitrary tasks in ROBOSIM.

3. We introduce INSTALIGN, an *instruction alignment* procedure to refine instruction-code pairs to improve alignment between instructions and code generated by SELF-INSTRUCT.

4. We fine-tune a small open-weight model, Codellama-Python-7B [30], using ROBO-INSTRUCT, and improve its performance to outperform several CodeLLMs, including Deepseek-Coder-33B [8], and Starcoder2-15B [21] and two proprietary LLMs, GPT-3.5-Turbo [27] and Gemini-1.0-Pro [33] on the ROBOEVAL benchmark.

Our code and data will be released at URL `anonymized`.

## 2 ROBO-INSTRUCT

In this section, we present how ROBO-INSTRUCT generates training datasets of domain-specific robot programs. Alg. 1 shows a broad overview of the framework. To add an entry in the training dataset, SELF-INSTRUCT first generates an instruction-program pair, $(\mathcal{I}, \mathcal{P})$, from the robot APIs and seed tasks, shown in Appendix A.4. Then, ROBOSIM dynamically synthesizes a *consistent* world state *on the fly* as it executes and validates $\mathcal{P}$. If $\mathcal{P}$ is invalid, ROBO-INSTRUCT employs a rejection-sampling method, which generates a new program $\mathcal{P}$ given the same $\mathcal{I}$ and evaluates the new $\mathcal{P}$ again. This process repeats until $\mathcal{P}$ becomes valid or a predefined maximum resampling limit is reached. If the limit is reached, the instruction might be invalid given the domain-specific APIs or too complex to generate a program, so the instruction-program pair is discarded. Finally, if $\mathcal{P}$ is valid, INSTALIGN takes in $(\mathcal{I}, \mathcal{P})$ to revise $\mathcal{I}$ to better reflect the intent of $\mathcal{P}$ and the aligned instruction and program is saved to the training dataset. In the following subsections, we elaborate on the specific design of each component.

---

**Algorithm 1** ROBO-INSTRUCT: Instruction-Program Generation

---

**Require:** $\mathcal{S}$,        ▷ Robot API and seed tasks,
     **Let** $\mathcal{P} \leftarrow$ Program,        ▷ The program begin checked
     **Let** $\mathcal{I} \leftarrow$ Instruction,        ▷ The instruction corresponding to $\mathcal{P}$
     **Let** ROBOSIM: $\mathcal{P} \rightarrow$ bool,        ▷ Domain-specific task-agnostic simulator
     **Let** INSTALIGN: $\mathcal{S} \times \mathcal{I} \times \mathcal{P} \rightarrow \mathcal{I}$,        ▷ Instruction-program alignment model
     **Let** SELF-INSTRUCT$_{inst}$: $\mathcal{S} \rightarrow \mathcal{I}$,        ▷ SELF-INSTRUCT instruction generation model
     **Let** SELF-INSTRUCT$_{code}$: $\mathcal{S} \times \mathcal{I} \rightarrow \mathcal{P}$,        ▷ SELF-INSTRUCT program generation model
1: **Initialize:** $\mathcal{D} = \emptyset$        ▷ Training dataset
2: **Initialize:** $N$        ▷ Training dataset size
3: **Initialize:** $m$        ▷ Maximum resampling limit
4: **while** len($\mathcal{D}$) $< N$ **do**
5:     $\mathcal{I} \leftarrow$ SELF-INSTRUCT$_{inst}(\mathcal{S})$
6:     $\mathcal{P} \leftarrow$ SELF-INSTRUCT$_{code}(\mathcal{S}, \mathcal{I})$
7:     **for** $i = 1$ **to** $m$ **do**
8:        is_program_valid = ROBOSIM($\mathcal{P}$)        ▷ Validate the program
9:        **if** is_program_valid = FALSE **then**
10:          $\mathcal{P} \leftarrow$ SELF-INSTRUCT$_{code}(\mathcal{S}, \mathcal{I})$        ▷ Rejection-sampling
11:        **else**
12:          $\mathcal{I}_{\text{aligned}} \leftarrow$ INSTALIGN($\mathcal{S}, \mathcal{I}, \mathcal{P}$)        ▷ Align instruction with the program
13:          $\mathcal{D} \leftarrow (\mathcal{I}_{\text{aligned}}, \mathcal{P})$
14:          **break**
15:        **end if**
16:     **end for**
17: **end while**
18: **return** $\mathcal{D}$

---

## 2.1 ROBOSIM: A Task-Agnostic Simulator For Domain-Specific Programs

We present a principled approach to design ROBOSIM for validating domain-specific robot programs. Alg. 2 illustrates the high-level algorithm used to assess the correctness of a robot program. ROBOSIM employs the concept of *world state* to simulate the robot actions directed by a program, ensuring consistent and reliable evaluation. A world state is a symbolic representation of the environment in which the robot operates, and it keeps track of the high-level changes in the robot state and the surrounding environment as the robot performs actions in order. For example, consider a program instruction that commands a robot to check if an apple is nearby. The world state queries the stored information about the surrounding environment, identifies all objects at the robot's current location, and informs the program whether an apple is present.

However, since SELF-INSTRUCT generates arbitrary programs based on the provided APIs, ROBOSIM does not know what a plausible world state relevant to the program would be a priori — *e.g.,* reasoning about the existence of an apple in the example program. Thus, we equip ROBOSIM with the ability to expand the world state as more robot actions are performed. Our approach is inspired by angelic execution [4], which has previously been used for software verification of programs with partially defined library functions. In our case, instead of partially defined library functions, we have unknown plausible world states. ROBOSIM *dynamically* synthesizes and grows a world state based on domain-specific constraints (*e.g.,* object permanence, robot skills, *etc.*) and the execution trace of the program, which allows it to infer a consistent and relevant world state.

Specifically, ROBOSIM modifies the program to replace all API calls with the DYNAMICEVAL function (Alg. 2 line 4) — when an API function is called during execution, the DYNAMICEVAL function is invoked instead.

DYNAMICEVAL makes an important extension to the formulation of STRIPS [7] to integrate with API functions. DYNAMICEVAL equips each API function with specific pre-conditions, effects, and return values. The pre-conditions are composed of literals tailored to the function's requirements. For instance, the API function `is_in_room('apple')`, which determines if an object 'apple' is in the same room as the robot, uses two literals for its pre-condition: `robot_at(X)` and `obj_at(X, 'apple')`. Generally, STRIPS assigns one of two possible values to each literal: `True` if the literal is defined, otherwise `False`. However, prior to program execution, DYNAMICEVAL is unaware of the program-relevant literals. Thus we assign a third value, *undefined*, to such unknown literals. Literals must thus be explicitly defined as either `True` or `False`, or they remain undefined if not specified.

Alg. 3 demonstrates how DYNAMICEVAL executes an API function and updates the world state. First, it calculates the precondition specified for the function. It then checks each literal in the precondition to see if it is defined. If a literal is undefined, DYNAMICEVAL invokes GROWWORLD, a stochastic function that assigns a random truth value to the literal and updates the world state accordingly. Finally, DYNAMICEVAL proceeds to execute the API function using the current world state, retrieves the return values, and applies the function's effects to update the world state.

Fig. 2 illustrates an example of ROBOSIM executing a generated program. Initially, ROBOSIM's world state only specifies the robot's current location, and whether a pie is in the same room as the robot remains undefined (line 2). Therefore, DYNAMICEVAL invokes GROWWORLD to

---

**Algorithm 2** ROBOSIM($\mathcal{P}$)

**Require:** Program $\mathcal{P}$                          ▷ Generated program

1: **Initialize:** Set $\mathcal{A}$            ▷ A set of domain-specific robot APIs
2: **Initialize:** $k$            ▷ Number of evaluation iterations
3: **Initialize:** $\mathcal{W}_{\text{init}}$       ▷ An initial world state with or without predefined information
4: $\mathcal{P}_{\text{trans}} \leftarrow$ TRANSLATE$(\mathcal{P}, \mathcal{A}, $DYNAMICEVAL$)$      ▷ Replace each API call with DYNAMICEVAL
5: **for** $i = 1$ **to** $k$ **do**           ▷ Then, evaluate $\mathcal{P}$ $k$ times to catch program errors
6:     **try:**
7:        $\mathcal{W} \leftarrow \mathcal{W}_{\text{init}}$           ▷ Initialize a new world state
8:        `exec`$(\mathcal{P}_{\text{trans}}, \mathcal{W})$
9:     **catch:**
10:        **return** False
11: **end for**
12: **return** True          ▷ Return True if all program executions are successful

---

**Algorithm 3** DYNAMICEVAL(api_fn, params, $\mathcal{W}$)

```
1:  p ← GETPRECOND(api_fn, params)        ▷ Get the parameter-specific precondition for api_fn
2:  for l ∈ p do                          ▷ Loop through every literal in the precondition
3:    if CHECKDEFINED(W, l) == undefined then
4:      W ← GROWWORLD(l, W)               ▷ Instantiate the literal and grow W to include it
5:    end if
6:  end for
7:  retval, W ← EXECUPDATE(api_fn, params, W)          ▷ Execute api_fn and update W
8:  return retval, W
```

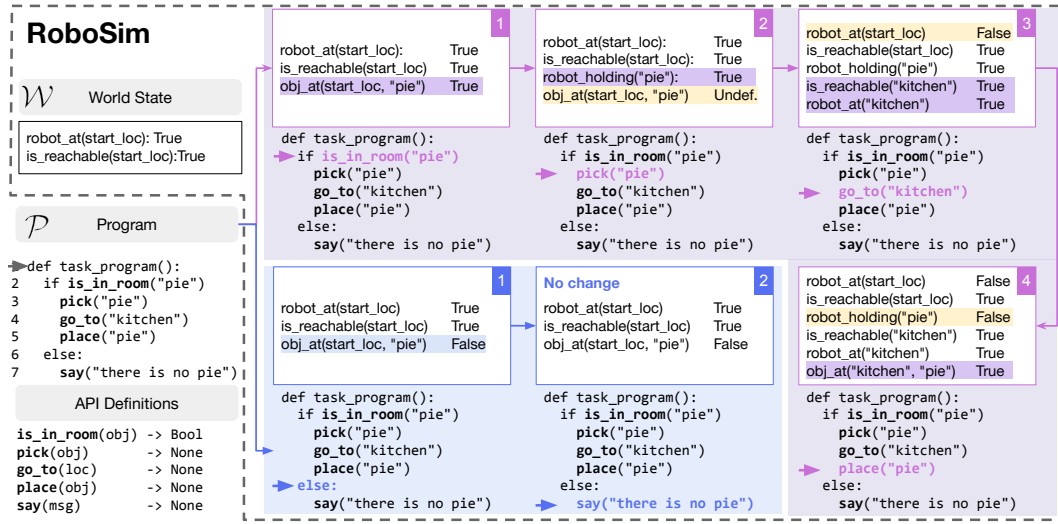

Figure 2: Example of ROBOSIM executing a generated program and updating the world state. Initially, ROBOSIM begins with a world state that includes only the robot's current location. As the program executes, two distinct execution paths emerge, depicted in light purple and blue. This figure demonstrates how the world state is updated along each execution path.

randomly determine a truth value for the `obj_at(start_loc, "pie")` literal, leading to two distinct execution paths depicted in light purple and blue. Subsequently, as additional API functions are called, more literals are introduced or updated in the world state to ensure consistent evaluations.

Finally, due to the stochastic nature of DYNAMICEVAL, ROBOSIM must execute the generated program multiple times to validate the program. If all executions are successful, the program is deemed correct (Alg. 2 line 5-11).

## 2.2  INSTALIGN: Instruction-Program Alignment Procedure

Given that LLMs are extensively trained in code understanding [30], INSTALIGN is a procedure that prompts an LLM to revise $\mathcal{I}$ to better reflect the intent of $\mathcal{P}$. This procedure involves two steps: first, given $\mathcal{I}$ and $\mathcal{P}$, INSTALIGN leverages Chain-of-Thought reasoning [37] (CoT) to prompt an LLM to generate a revised instruction, $\mathcal{I}_{\text{revised}}$; second, INSTALIGN invokes the LLM again to determine whether $\mathcal{I}$ or $\mathcal{I}_{\text{revised}}$ is more aligned with $\mathcal{P}$'s intent and output the chosen instruction as $\mathcal{I}_{\text{aligned}}$.

To generate $\mathcal{I}_{\text{revised}}$, the prompt to the LLM comprises the robot API function definitions, $\mathcal{I}$, $\mathcal{P}$, and CoT instructions. The CoT asks the LLM to perform the following three steps in order: 1. write down all the robot APIs used in the program; 2. examine these APIs and write down step by step what the program does; 3. combine all the information above to revise the robot instruction. Similarly, to determine $\mathcal{I}_{\text{aligned}}$, an LLM is prompted to think step by step about $\mathcal{P}$, $\mathcal{I}$ and $\mathcal{I}_{\text{revised}}$ to arrive at a conclusion. Detailed prompt is shown in Appendix A.6.

# 3  Analysis and Experiments

In this section, we investigate the following two research questions:

1. Is ROBO-INSTRUCT effective at generating training data to fine-tune a small language model for generating domain-specific robot programs?
2. How do ROBOSIM and InstAlign impact the effectiveness of ROBO-INSTRUCT?

We conduct our investigation by fine-tuning the Codellama-Python-7B model [30] on the synthetic dataset generated by ROBO-INSTRUCT and evaluate the fine-tuned model using ROBOEVAL [10], a domain-specific code generation benchmark for service mobile robots. In the following subsections, we first provide a brief description of ROBOEVAL. Then we present our experimental results addressing the two main research questions. Finally, we offer more analysis of ROBOSIM, INSTALIGN, and the synthetic dataset.

## 3.1 ROBOEVAL: A Domain-Specific Robot Code Generation Benchmark

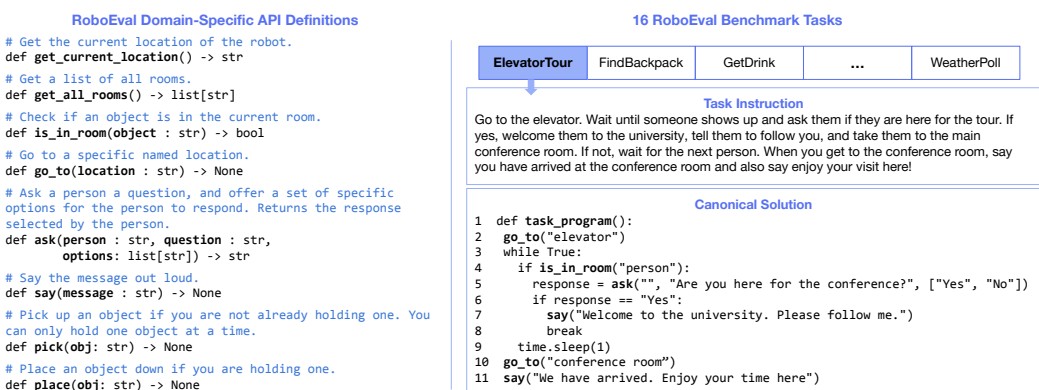

Figure 3: ROBOEVAL APIs and benchmark task example.

ROBOEVAL is a domain-specific code generation benchmark, featuring a suite of 16 tasks designed to evaluate the ability of LLMs to understand custom APIs and generate programs for service robots. In this domain, a service robot can perceive objects, navigate to various locations, manipulate items, and communicate with humans. Furthermore, the robot should be capable of basic commonsense reasoning and executing complex tasks that involve conditional and repetitive actions. To facilitate these capabilities, ROBOEVAL defines a set of 8 API functions in Python as skill primitives. Fig. 3 illustrates these function signatures and definitions, alongside an example task instruction and its canonical solution from the benchmark. In addition, unlike other popular code generation benchmark tasks [2, 6, 9, 14, 16, 19], *the order of the robot's actions is crucial for successfully completing the specified tasks*. For instance, in the task *"bring me a marker from the classroom that does not have a whiteboard,"* the robot must check each classroom until it finds one without a whiteboard, whereas simply bringing back a marker is insufficient. Hence, ROBOEVAL evaluates the generated program by executing it in a simulator to capture the action traces, which are subsequently validated for sequence correctness using temporal logic.

## 3.2 RQ1: Is ROBO-INSTRUCT Effective at Generating Training Data to Fine-Tune a Small Language Model for Generating Domain-Specific Robot Programs?

**Experiment Setup.** We use the open-weight LLM, Llama3-8B-Inst, for ROBO-INSTRUCT. To generate a diverse dataset, we employ nucleus sampling for creating instruction-program pairs, setting the temperature $T = 1$ and top $p = 0.95$. The maximum resampling limit is capped at 3 to accommodate instructions that initially produce invalid programs. For the LLM used in INSTALIGN, we empirically adjust the generation temperature to $T = 0.3$ to optimize performance. Furthermore, we assess the edit similarity between token sequences of each instruction pair in the dataset [15], removing duplicates where the similarity score exceeds 0.6. We use the same setup to generate data via SELF-INSTRUCT. Instead of discarding invalid programs, SELF-INSTRUCT includes every generated instruction-program pair in the training dataset. Finally, we create two datasets with 5K instruction-program pairs each using SELF-INSTRUCT and ROBO-INSTRUCT respectively. These datasets are then used to fine-tune the Codellama-Python-7B model. The learning rate is set to be

| Fine-tune | Model | # Param | ROBOEVAL pass@1 | | Licensing |
|---|---|---|---|---|---|
| | | | $T = 0$ | $T = 0.2$ | |
| - | GPT-4 | - | **83.75%** | **85.81%** | Proprietary |
| - | GPT-3.5 | - | 67.5% | 65.56% | Proprietary |
| - | Gemini-1.0-Pro | - | 60.00% | 59.88% | Proprietary |
| - | Codellama-Python | 7B | 40.00% | 39.31% | Open |
| - | Codellama-Python | 34B | 46.25% | 48.25% | Open |
| - | Starcoder2 | 15B | 62.5% | 60.94% | Open |
| - | Deepseek-Coder | 33B | 53.75% | 52.13% | Open |
| - | Llama3-Inst | 8B | 48.75% | 48.38% | Open |
| Self-Instruct | Codellama-Python | 7B | 55.00% | 52.69% | Open |
| Robo-Instruct (ours) | Codellama-Python | 7B | **68.75%** | **66.00%** | Open |

Table 1: Pass@1 results of different LLMs on ROBOEVAL computed with greedy decoding $T = 0$ and nucleus sampling $T = 0.2$.

$3e$-5 with a warmup ratio of $3\%$ and a constant lr scheduler. We employ the AdamW optimizer [20] with an effective batch size of 8, training each model for 5 epochs using a sequence length of 2048 tokens. We train all our models on a single H-100 GPU using unsloth [35].

**Baselines.** We divide our baseline models into 2 categories: 1) proprietary LLMs, including GPT4 [28], GPT3.5-Turbo [27], Gemino-Pro [33], and 2) open-weight LLMs, including Codellama-Python-7B [30], Codellama-Python-34B, Starcoder2-33B [21], Deepseek-Coder-33B [8], and Llama3-8B-Inst [1]. All the results are evaluated using ROBOEVAL and reported in Tab. 1.

Tab. 1 presents the average pass@1 results for different LLMs on ROBOEVAL, using two different temperature settings for generation: greedy decoding at a temperatures of $T = 0$ and nucleus sampling at a temperature of $T = 0.2$. The results show that ROBO-INSTRUCT-fine-tuned Codellama significantly improves upon the base Codellama-Python-7B and outperforms the SELF-INSTRUCT-fine-tuned variant. Notably, it surpasses all open-weight models, including larger ones like Codellama-Python-34B and Deepseek-Coder-33B. Additionally, although the training dataset was generated using Llama3-8B-Inst, which scores less than 50% pass@1 on ROBOEVAL, our ROBO-INSTRUCT-fine-tuned model still achieves a significant improvement, scoring 68.75% under deterministic temperature settings for generation. Finally, compared to proprietary models, while our ROBO-INSTRUCT-fine-tuned model trails the more powerful GPT-4, it outperforms GPT-3.5-Turbo and Gemini-1.0-Pro in generating programs for service mobile robots. This result demonstrates the effectiveness of our approach in generating domain-specific robot program data for fine-tuning a small language model. It suggests that the fine-tuned model could potentially replace some proprietary models, providing a more cost-effective and private option for local deployment.

### 3.3 RQ2: How Do ROBOSIM and InstAlign Impact the Effectiveness of ROBO-INSTRUCT?

| Method | T=0 | | T=0.2 | | Invalid Programs |
|---|---|---|---|---|---|
| | pass@1 | Improv. | pass@1 | Improv. | |
| Codellama-7B-Python | 40.00% | +0% | 39.31% | +0% | 38.31% |
| SELF-INSTRUCT | 55.00% | +15.00% | 52.69% | +13.38% | 20.94% |
| +Reject Unsolvable (RU) | 60.00% | +20.00% | 57.62% | +18.31% | 23.38% |
| +ROBOSIM + RU | 63.75% | +23.75% | 63.88% | +24.57% | **14.13%** |
| +INSTALIGN + RU | 58.75% | +18.75% | 59.81% | +20.50% | 23.44% |
| +Both (ROBO-INSTRUCT) | **68.75%** | **+28.75%** | **66.00%** | **+26.69%** | 17.07% |

Table 2: Pass@1 results of different LLMs on ROBOEVAL computed with greedy decoding $T = 0$ and nucleus sampling $T = 0.2$.

Using the same setup as in the previous section, we investigate the effectiveness of ROBOSIM and INSTALIGN. Since SELF-INSTRUCT may generate invalid instructions that no corresponding valid program can pass in ROBOSIM, we propose rejecting these unsolvable instructions (we name

this process RU) to evaluate the upperbound performance of SELF-INSTRUCT. Tab. 2 shows the average pass@1 results from Codellama-7B-Python fine-tuned on different datasets generated by each method. First, findings from SELF-INSTRUCT + RU indicate that simply discarding invalid instructions could also improve model performance. Additionally, fine-tuning with a dataset created from SELF-INSTRUCT+RoboSim results in the smallest proportion of invalid program errors. Finally, while incorporating either ROBOSIM or INSTALIGN individually offers some improvement over the baseline SELF-INSTRUCT + RU results, ROBO-INSTRUCT still results in the best performance. This indicates that the integration of these two components is important to the framework's effectiveness.

## 3.4 Qualitative analysis of the generated program errors

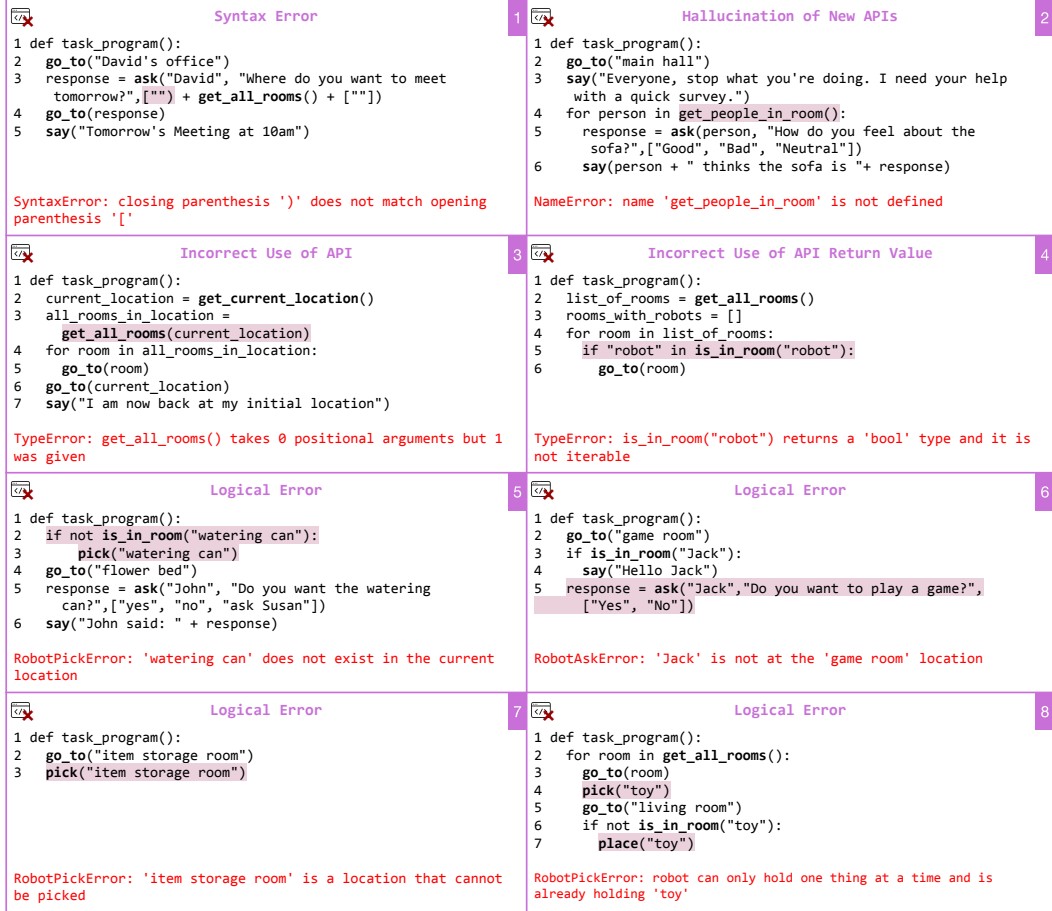

Figure 4: SELF-INSTRUCT-Generated Program Errors: Examples 1 to 4 illustrate errors specific to the Python language, and Examples 5 to 8 highlight errors rooted in domain-specific constraints.[2]

We analyze invalid programs identified by ROBOSIM, categorizing the errors into two types: language-native errors and domain-specific constraint violations. Fig. 4 displays eight examples of these programs, with Examples 1 to 4 illustrating errors specific to the Python language, and Examples 5 to 8 highlighting errors rooted in domain-specific constraints. Language-native errors are generally straightforward, such as syntax errors, the use of undefined variables or functions, or improper use of provided APIs.

In contrast, errors related to domain-specific constraints tend to be more complex to detect. For instance, Example 5 illustrates the program incorrectly trying to pick up a watering can (line 3) after establishing that it is not present at the location (line 2). Similarly, Example 6 demonstrates an error where the program inappropriately asks Jack (line 5) after confirming his absence from the room

---

[2]Programs have been adapted to succinctly demonstrate the types of errors and fit within the figure.

(line 3). Example 7 illustrates a scenario in which ROBOSIM updates the world state by labeling "item storage room" as a location after executing the `go_to` command (line 2). Subsequently, the robot attempts to pick up this location (line 3), resulting in an error. Example 9 is the most intricate scenario where the world state in the living room is updated to include a toy after the robot places it there (line 7). When the robot returns to the living room for the second time (line 5), it does not place down what it holds (line 7). Hence, in the third room the robot visits (line 3), when it attempts to pick up a toy again (line 4), an error occurs because the robot can only carry one item at a time.

## 4   Related Work

### 4.1   LLMs for Robot Code Generation

LLMs have shown impressive capabilities in generating robot programs from natural language [11, 17, 31]. One popular approach uses LLMs to generate composable costmaps for robots to plan their motion on. In this approach, Voxposer [12] focuses on the tabletop manipulation setting and NavCon [3] focuses on creating composable maps for navigation. Using LLM to create reward functions is also promising. Eureka [23, 24] and Language to Rewards for Robotic Skill Synthesis [41] both show that LLM can generate good reward functions that allows robots to acquire complex skills. Finally, LLM can also be used to generate programs for high-level planning. LLM+p [18] outputs a robot plan in the form of the well-defined planning domain definition language (PDDL). Tidybot [39] uses an LLM to generate a rule that captures user preferences from examples and executes a program to sequentially complete the task in order. RoboEval [10] focuses on generating domain-specific programs for service mobile robots. It generates a program that allows the service robot to carry out long-horizon tasks and then validates the correctness of the program.

### 4.2   Generating Datasets For Fine-tuning LLMs

To enhance LLMs' performance in code generation, numerous studies have explored the creation of specialized datasets [13, 25, 26]. SELF-INSTRUCT [36] is one popular method for generating synthetic datasets using an LLM. Following this methodology, Alpaca [32] generates 52K instruction-following demonstrations and subsequently fine-tunes the LLaMA 7B model [34] to create Alpaca 7B, which can behave qualitatively similarly to OpenAI's text-davinci-003. Code Alpaca [5] extends this approach to generate code instructions using 21 seed tasks, while Gorilla-LM [29] adapts the method to focus on ML domain-specific APIs from Huggingface, TensorFlow Hub, and Torch Hub. To create more complex instructions, Evol-Instruct [22, 40] proposes iteratively updating instructions to become more complex through different prompting strategies. In addition to Evol-Instruct, OSS-Instruct [38] uses open-source code snippets to generate 75K high-quality instruction data and fine-tunes the Codelllama-Python-7B model to create Magicoder, which can match the performance of GPT-3.5-Turbo [27] on HumanEval [6]. While these works focus on creating seed instruction sets to generate synthetic data for effectively fine-tuning an LLM, our research investigates post-processing methods in addition to SELF-INSTRUCT. Specifically, we concentrate on generating domain-specific programs in robotics [10], where we can effectively leverage constraints to filter out erroneous programs.

## 5   Conclusion, Limitation and Future Works

In this work, we introduce ROBO-INSTRUCT, a novel framework to generate synthetic training data to fine-tune small language models for domain-specific robot programs. ROBO-INSTRUCT comprises two novel components: 1) ROBOSIM, an angelic-execution-based algorithm to effectively validate SELF-INSTRUCT-generated programs, and 2) INSTALIGN, an instruction alignment procedure to revise instructions to better align with the generated programs. The experimental results demonstrate that the Codellama-Python-7B model fine-tuned on the ROBO-INSTRUCT-generated dataset can significantly outperform many popular open-weight LLMs for generating domain-specific robot programs. It also outperforms two proprietary LLMs, GPT-3.5-Turbo and Gemino-1.0-Pro, as well as the SELF-INSTRUCT-fine-tuned variant. A limitation of this study is that ROBO-INSTRUCT relies on SELF-INSTRUCT to filter invalid programs, making the dataset quality dependent on SELF-INSTRUCT's performance. This can introduce biases if SELF-INSTRUCT consistently fails in certain areas. Future work will explore integrating ROBO-INSTRUCT with advanced methods like Evol-Inst and OSS-Inst to enhance dataset quality for domain-specific robot programs.

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

# A Appendix

## A.1 Overview

In this appendix, we first present ablation experiments to investigate the percentage of invalid programs generated by SELF-INSTRUCT and examine how the generation temperature in INSTALIGN affects final performance. Next, we analyze and compare the datasets generated by ROBO-INSTRUCT and SELF-INSTRUCT. Finally, we list the seed tasks used in ROBOEVALand the CoT prompt.

## A.2 Ablation Exmperiments

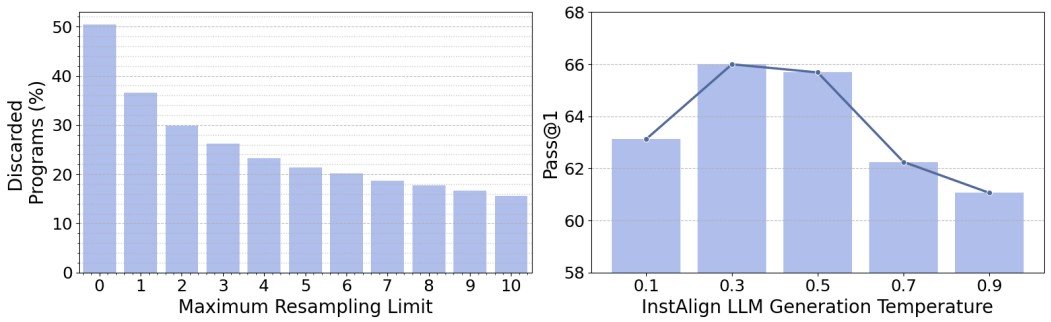

Figure 5: Ablation Experiments

### A.2.1 effectivenss of the simulator

We analyze the percentage of instruction-program pairs discarded by ROBOSIM at various maximum resampling limits, as shown in Fig. 5. Initially, with the maximum resampling limit set to 0, disabling the rejection-sampling method, approximately 51% of the programs generated by SELF-INSTRUCT contain errors. As the limit increases, fewer programs are discarded. However, there is a diminishing return; even with the maximum resampling limit set to 10, about 15% of the instructions still result in invalid programs.

### A.2.2 Instruction Alignment model temperature

We further investigate how varying LLM temperatures for generating $\mathcal{I}_{\text{revised}}$ in INSTALIGN impact the performance of the fine-tuned model. Fig. 5 shows the bar chart of the pass@1 score of the models fine-tuned over datasets generated using different LLM temperatures in INSTALIGN. The model performs the best when fine-tuned on the dataset generated using LLM temperature $T = 0.3$. As the temperature increases, we observe a decrease in performance.

## A.3 Analysis of the Generated Datasets

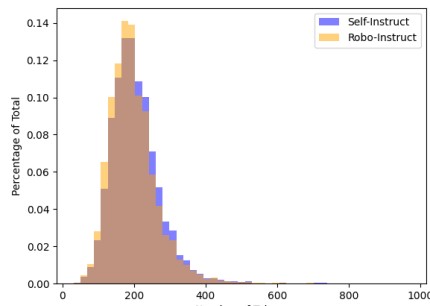
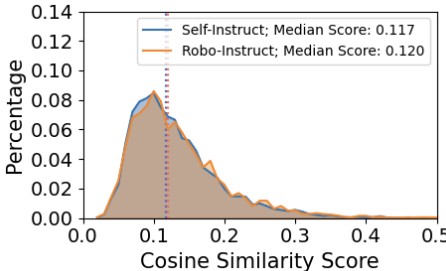

(a) Token Length Distribution for SELF-INSTRUCT *vs.* ROBO-INSTRUCT

(b) Cosine Similarity with ROBOEVALfor SELF-INSTRUCT *vs.* ROBO-INSTRUCT

Figure 6: Dataset Analysis

| Method | Size | Ngram=4 Score | # Synth. Loc. | # Synth. Obj. |
|---|---|---|---|---|
| ROBO-INSTRUCT | 5K | 0.587 | 1025 | 928 |
| SELF-INSTRUCT | 5K | 0.581 | 956 | 1060 |

Table 3: Dataset Statistics

We first compute and plot the distribution of token lengths in the SELF-INSTRUCT-generated dataset and the ROBO-INSTRUCT-generated dataset, as shown in Fig. 6(a). Next, we measure the cosine similarity between each dataset and the ROBOEVALbenchmark tasks following the approach in Magicoder [38], as depicted in Fig. 6(b). Finally, Tab. 3 presents the n-gram diversity score of each dataset, along with the number of synthesized locations and objects. Our findings indicate that both distributions and dataset statistics are very similar, suggesting that ROBO-INSTRUCT enhances the quality of the generated data over SELF-INSTRUCT rather than merely aligning the dataset towards the benchmark tasks.

## A.4 ROBOEVALSeed Task Example

```python
# Instruction: Go to Arjun's office,
# ask him if he is ready to head out,
# and come back and tell me what he said
def task_program():
    start_loc = get_current_location()
    go_to("Arjun's office")
    response = ask("Arjun",
        "Are you ready to go?",
        ["Yes", "No"])
    go_to(start_loc)
    say("Arjun said: " + response)
```

Listing 1: Seed Task Example 1

```python
# Instruction: Ask Alice if she needs 1, 2, or 3 boxes.
# Go to the storage room and ask if they have that many boxes.
# If so, go place the boxes in Alice's office.
# Otherwise, tell Alice you could not get the boxes.
def task_program():
    go_to("Alice's office")
    num_boxes = ask("Alice",
        "How many boxes do you need?",
        ["1", "2", "3"])
    go_to("storage room")
    response = ask("",
        "Do you have" + num_boxes + " boxes?",
        ["Yes", "No"])
```

```
512     if response == "Yes":
513         for _ in range(int(num_boxes)):
514             pick("box")
515             go_to("Alice's office")
516             place("box")
517             go_to("storage room")
518     else:
519         go_to("Alice's office")
520         say("I could not get the boxes")
```

Listing 2: Seed Task Example 2

```
521 # Instruction: Check if there is a red marker in the main
522 # office, and if so, tell Eve that there is a marker there.
523 # If not, go to the supply room and
524 # bring a red marker to the main office.
525 def task_program():
526     go_to("main office")
527     red_marker_found = is_in_room("red marker")
528     if red_marker_found:
529         go_to("Eve's office")
530         say("There is a red marker in the main office")
531     else:
532         go_to("supply room")
533         pick("red marker")
534         go_to("main office")
535         place("red marker")
```

Listing 3: Seed Task Example 3

```
536 # Instruction: Check every classroom if there is a whiteboard.
537 # Go to Aiden's office to tell him which room does not
538 # have a whiteboard. Come back and tell me task is completed.
539 def task_program():
540     start_loc = get_current_location()
541     list_of_rooms = get_all_rooms()
542     room_without_whiteboard = []
543     for room in list_of_rooms:
544         if "classroom" not in room:
545             continue
546         go_to(room)
547         if not is_in_room("whiteboard"):
548             room_without_whiteboard.append(room)
549     go_to("Aiden's office")
550     if len(room_without_whiteboard) > 0:
551         message = ""
552         for room in room_without_whiteboard:
553             message += room + ", "
554         message += "do not have a whiteboard"
555     else:
556         message = "all classrooms have a whiteboard"
557     say(message)
558     go_to(start_loc)
559     say("task is completed")
```

Listing 4: Seed Task Example 4

```
560 # Instruction: Go to the kitchen and wait for someone
561 # to show up. When someone shows up, ask them to open
562 # the fridge, then pick up a diet coke.
563 # Finally, put the diet coke in the living room.
564 def task_program():
565     go_to("kitchen")
566     while True:
567         if is_in_room("person"):
```

```
568          response = ask("",
569              "Please open the fridge",
570              ["Yes", "No"])
571          if response == "Yes":
572              pick("diet coke")
573              break
574      time.sleep(1)
575  go_to("living room")
576  place("diet coke")
```

Listing 5: Seed Task Example 5

```
577  # Instruction: Take a bed sheet from the laundry room
578  # and put it in each of the bedrooms.
579  def task_program():
580      start_loc = get_current_location()
581      list_of_rooms = get_all_rooms()
582      for room in list_of_rooms:
583          if "bedroom" not in room:
584              continue
585          go_to("laundry room")
586          pick("bed sheet")
587          go_to(room)
588          place("bed sheet")
589      go_to(start_loc)
```

Listing 6: Seed Task Example 6

## A.5 Prompts to Generate Synthetic Dataset Using SELF-INSTRUCT

---

You are a helpful assistant. Here is a robot that has the following capabilities:
- def get_current_location() -> str:
- def get_all_rooms() -> list[str]:
- def is_in_room(object : str) -> bool:
- def go_to(location : str) -> None:
- def ask(person : str, question : str, options: list[str]) -> str:
- def say(message : str) -> None:
- def pick(obj: str) -> None:
- def place(obj: str) -> None:
Generate an interesting robot task that can be accomplished using the above capabilities.
{{SEED EXAMPLE}}

Generate an interesting robot task that can be accomplished using the above capabilities.
...

---

Table 4: Prompts to Generate Synthetic Dataset Using SELF-INSTRUCT.

## A.6 CoT Prompts for INSTALIGN

### Role: You are an expert at understanding robot programs. You will be given a task instruction and robot program pair. However, the instruction may not align with the program well. You need to correct the task instruction to match the given robot program.

### Context: The robot only has access to the following 8 APIs and standard Python functions
- def get_current_location() -> str:
- def get_all_rooms() -> list[str]:
- def is_in_room(object : str) -> bool:
- def go_to(location : str) -> None:
- ask(person : str, question : str, options: list[str]) -> str:
- say(message : str) -> None:
- def pick(obj: str) -> None:
- def place(obj: str) -> None:

### Inputs
Original Instruction: This is a task instruction that may not align with the robot program Robot Program: This is a python function starting with 'def task_program():'

### Task:
1. Write down all the provided APIs used in the program and explain the effect of each API in this program
2. Examine these APIs and write down step by step what the program does
3. Combine all the results above and rewrite the instruction under # Final Corrected Instruction: You need to be specific and clear in your final corrected instruction.

Table 5: CoT Prompts for INSTALIGN.

## B CheckList

1. **[Claims]** Yes. The research questions listed in the evaluation section are formulated so as to directly reflect the claims of the paper.

2. **[Limitations]** Yes. This is discussed in Section 5.

3. **[Theory, Assumptions and Proofs]** N/A. We do not have any theoretical results.

4. **[Experimental Result Reproducibility]** Yes. We provide the training hyperparameters in Section 4. We will also release our model upon acceptance.

5. **[Open Access to Data and Code]** Yes. We provide the prompts that are used to generate the training dataset. We will also release our training dataset upon acceptance.

6. **[Experimental Setting/ Details]** Yes. We discuss the details of the training scheme in Section 3.2, which follows the standard approach to fine-tuning an LLM.

7. **[Experiment Statistical Significance]** Yes. We performed ablation studies to validate our methods in Section 3.3.

8. **[Experiments Compute Resource]** Yes. We mention that we train all our models on a single H-100 GPU using unsloth in Section 3.2.

9. **[Code Of Ethics]** Yes

10. **[Broader Impacts]** N/A: This paper addresses an existing problem (using LLMs to synthesize robot programs [10, 12, 17]), and does not introduce any novel concerns beyond the existing scope.

11. **[Safeguards]** N/A: the programs we will generate or release are domain-specific with respect to RoboEval [10], which has existing safeguards in place.

12. **[Licenses]** Yes — we build on SELF-INSTRUCT, Llamav3 [30], and RoboEval [10] with attribution, and Table 1 refers to the licenses of the models used in the evaluation.

13. **[Assets]** N/A. The code we will release will include details of documentation, training, license, and limitations. The code will be released upon acceptance.

14. **[Crowdsourcing and Research with Human Subjects]** N/A

15. **[IRB Approvals]** N/A

