# OpenReview forum: "Robo-Instruct: Simulator-Augmented Instruction Alignment For Finetuning CodeLLMs"
_NeurIPS.cc/2024/Conference — Submitted to NeurIPS 2024_

### Official Review · Reviewer_BHTn · 2024-07-12

**Soundness:** 3
**Presentation:** 3
**Contribution:** 3
**Rating:** 8
**Confidence:** 4

**Summary:**

The paper introduces ROBO-INSTRUCT, a novel framework designed to generate synthetic training data for fine-tuning small language models to create domain-specific robot programs. The framework features two main components: ROBOSIM, an algorithm that validates programs using angelic execution, and INSTALIGN, which aligns instructions with the generated programs.
The key contributions of this work include the development of ROBO-INSTRUCT to enhance the code generation performance of small open-weight language models for domain-specific robot programs. This framework introduces ROBOSIM, which features a dynamic world synthesis and evaluation process for generating relevant world states and performing automated code checks for diverse tasks. Additionally, it includes INSTALIGN, a procedure that refines instruction-code pairs to improve alignment between instructions and the code generated by SELF-INSTRUCT. By fine-tuning the Codellama-Python-7B model using ROBO-INSTRUCT, the model significantly outperforms several other open-source and most proprietary models.

**Strengths:**

- The paper demonstrates strong clarity and organization, making complex concepts accessible to readers. Each section flows logically, and technical details are explained effectively, ensuring that the methodology and findings are easy to follow.
- The paper thoroughly reviews and incorporates current literature.
- The paper meticulously validates all its claims through experimental results and detailed analysis. The effectiveness of ROBO-INSTRUCT, ROBOSIM, and INSTALIGN is demonstrated convincingly through empirical data and comparisons with existing models and benchmarks. This empirical validation ensures that the contributions are not just theoretical but substantiated with practical evidence.

**Weaknesses:**

- While ROBO-INSTRUCT offers significant advancements for fine-tuning language models in robot programming, there are some weaknesses to consider. Firstly, it heavily relies on SELF-INSTRUCT for generating initial programs, potentially introducing biases from the base model's training data. This could limit the diversity and quality of the generated programs.
- Moreover, while ROBO-INSTRUCT shows promising results on benchmarks like ROBOEVAL, its application to real-world robot programming tasks requires thorough evaluation and validation. Real-world robot environments often present unpredictable challenges that benchmark datasets may not fully capture, necessitating further testing to assess the framework's robustness and generalizability in practical applications.

**Questions:**

What factors do you think contribute to GPT-4's superior performance compared to ROBO-INSTRUCT, despite your method showing competitive results against other models like GPT-3.5-Turbo and Gemini-1.0-Pro

**Limitations:**

The limitations of the work are clearly stated

---

> ### Author Rebuttal · Authors · 2024-08-06
>
> Thank you for your feedback. We would like to offer further clarification beyond the general response above to address the specific concerns and points raised in the review.
>
> ***Weakness:*** While ROBO-INSTRUCT offers significant advancements for fine-tuning language models in robot programming, there are some weaknesses to consider. Firstly, it heavily relies on SELF-INSTRUCT for generating initial programs, potentially introducing biases from the base model's training data. This could limit the diversity and quality of the generated programs.
>
> ***Response:***
>
> As discussed in Section 5 (Conclusion, Limitation, and Future Works), we acknowledge that our proposed method, which builds on Self-Instruct, could potentially be influenced by biases in the base model's training data. However, we would like to emphasize that our method is orthogonal to many other popular approaches used in Self-Instruct. Therefore, it is possible to apply Robo-Instruct alongside these methods to mitigate these concerns. For instance, the OSS-Instruct method by Magicoder (Wei et al., 2023) and the Evol-Instruct method by Wizardcoder (Luo et al., 2023) both aim to enhance the quality and diversity of data generated from Self-Instruct. Magicoder has demonstrated that combining these orthogonal methods (OSS-Instruct + Evol-Instruct) can lead to better outcomes. Consequently, we believe that using Robo-Instruct in conjunction with these methods could enhance the performance of fine-tuned models in domain-specific applications. Investigating the integration of these orthogonal methods for generating domain-specific datasets would be a valuable direction for future work.
>
> ***Weakness:*** Moreover, while ROBO-INSTRUCT shows promising results on benchmarks like ROBOEVAL, its application to real-world robot programming tasks requires thorough evaluation and validation. Real-world robot environments often present unpredictable challenges that benchmark datasets may not fully capture, necessitating further testing to assess the framework's robustness and generalizability in practical applications.
>
> ***Response:***
>
> We have deployed the Robo-Instruct fine-tuned model on a real robot using edge computing to demonstrate its real-world practicality, as illustrated in the PDF in the general response. This information will be included in the appendix of our revised paper.
>
> ***Question:*** What factors do you think contribute to GPT-4's superior performance compared to ROBO-INSTRUCT, despite your method showing competitive results against other models like GPT-3.5-Turbo and Gemini-1.0-Pro
>
> ***Response:***
>
> It is possible that GPT-4 has a significantly larger number of model parameters and is trained with more resources. As a result, GPT-4 has much longer inference times. In contrast, our fine-tuned model can perform inference five times faster, as illustrated in the PDF in the general response.

---

> > ### Comment · Area_Chair_nLBW · 2024-08-11
> > **Dear reviewer, please read and respond to authors' rebuttal.**
> >
> > This paper has very diverse reviews and it would benefit a lot to start a discussion to clarify any confusing points.
> >
> > Thanks!
> >
> > Your AC.

---

### Official Review · Reviewer_afBU · 2024-07-12

**Soundness:** 3
**Presentation:** 3
**Contribution:** 2
**Rating:** 4
**Confidence:** 3

**Summary:**

The paper tries to improve the performance of small open-sourced LLMs to generate code that can run successfully on a service robot simulator to solve tasks. The idea is to use another small model to generate program data using SELF-INSTRUCT and fine-tune a 7B model for the robotics domain. The authors note that data generated by SELF-INSTRUCT may have good diversity but lack correctness. To this end, they build a simulator RoboSIM that takes the programs generated by SELF-INSTRUCT and verifies the correctness of the execution in addition to syntax errors given a predefined robot API. They further modify the instructions to align with the verified programs better. Overall they show the 7B LLM fine-tuned on this clean data can outperform a GPT3.5-Turbo in the robotics domain.

**Strengths:**

The paper is well-written and motivated. Figures are easy to understand and helpful in conveying high-level ideas. The idea to build a pseudo-simulator that tracks world states without running the generated programs through an actual simulator is novel. It is amazing that synthetically generated data filtered by simple heuristics such as requiring programs to pass the world state tracking RoboSim is sufficient to help improve the performance of an actual simulator.

**Weaknesses:**

While the paper is well-written for the scope it sets for itself, I am not sure if the contribution is significant enough. There are many works using LLM to generate data and fine-tune domain-specific models, so the idea behind this paper is not super novel. The performance gain is also limited by the rather heuristic method considering the gap between the best model presented by the paper and GPT-4 it sought out to beat. In fact, given the 17% performance gap, simple baselines could be using GPT-4 to generate the programs and fine-tuning small models or using GPT-4 as the critic to filter programs. These simpler heuristics may yield better results and prove the Robo-Instruct method (which is also rather heuristic) proposed by the paper unnecessary. For reference consider
[1] Improving Small Language Models on PubMedQA via Generative Data Augmentation
Therefore, I am not sure the contribution of this paper is all that significant.

**Questions:**

1. Could you explain why InstAlign would improve pass@1 success rate by ~5% on top of RoboSIM+RU in table 2? What's the error bar/variance of these statistics? What is the intuition that InstAlign would help and could you give some examples?
2. One claim made early on in the paper is that Self-Instruct gives high-diversity data but lacks correctness, and a simulator can give correctness but lacks diversity. The paper is trying to get the best of both worlds. While the current experiment results show that Robo-Instruct does generate data with high correctness, we don't see evidence that Robo-Instruct preserves diversity. Can you show some evidence (through some metrics) on diversity?

**Limitations:**

Building RoboSim to track world states might work only for simple pick-n-place or task-planning problems. How to generalize this heuristic of tracking world states to more complex problems is unclear. Some analysis of the scope (suitable for what kind of problem class) of this approach is needed.

---

> ### Author Rebuttal · Authors · 2024-08-06
>
> Thank you for your feedback. We would like to offer further clarification beyond the general response above to address the specific concerns and points raised in the review.
>
> ***Weakness:*** While the paper is well-written for the scope it sets for itself, I am not sure if the contribution is significant enough. There are many works using LLM to generate data and fine-tune domain-specific models, so the idea behind this paper is not super novel.
>
> ***Response:***
>
> While fine-tuning domain-specific models using techniques such as Self-Instruct is common, obtaining high-quality correct instruction-program pairs remains a challenging problem. Since custom-defined domain-specific APIs may not be present in the training corpus of existing LLMs, methods like Self-Instruct can generate incorrect training data, as shown in Appendix A.2.1. Table 1 in the paper also highlights the performance gap when using the Self-Instruct method.
>
> Verifying the correctness of these generated programs using a simulator is a natural approach. However, creating an appropriate simulation environment for each program can be challenging, often requiring manual specifications of entities, types, and states. In this work, we address this challenge by introducing a novel approach that automates the generation of simulation environments for each generated program.  As outlined in the general response, this method is versatile, with the potential to be extensible beyond robotics, capable of handling arbitrary open-world tasks without the need for manual coding of entities, types, or states.
>
>
> ***Weakness:*** The performance gain is also limited by the rather heuristic method considering the gap between the best model presented by the paper and GPT-4 it sought out to beat. In fact, given the 17% performance gap, simple baselines could be using GPT-4 to generate the programs and fine-tuning small models or using GPT-4 as the critic to filter programs. These simpler heuristics may yield better results and prove the Robo-Instruct method (which is also rather heuristic) proposed by the paper unnecessary. For reference consider [1] Improving Small Language Models on PubMedQA via Generative Data Augmentation. Therefore, I am not sure the contribution of this paper is all that significant.
>
> ***Response:***
>
> Our preliminary results show that a simple distillation of GPT-4 is not effective for fine-tuning domain-specific models, and the best pass@1 score was around 64%, which is lower than the score reported in Robo-Instruct. We will add this information to the appendix of the revised paper.
>
> More importantly, we chose to generate data from an open-weight model instead of proprietary models like GPT-4 because open-weight models are often preferred in practice. They are cost-free, can be deployed on local servers for greater flexibility, and help address privacy concerns when dealing with sensitive information related to custom domain-specific APIs.
>
> ***Question:*** Could you explain why InstAlign would improve pass@1 success rate by ~5% on top of RoboSIM+RU in table 2? What's the error bar/variance of these statistics? What is the intuition that InstAlign would help and could you give some examples?
>
> ***Response:***
>
> The intuition behind why InstAlign would improve pass@1 accuracy is twofold:
> 1. ***Specificity of Aligned Instruction:*** The aligned instruction is more specific to the generated program, providing clearer guidance on the intended actions. Instead of offering generic instructions, it can communicate what the program aims to accomplish more precisely. For example:
> ```
> def task_program():
>     for person in ["Arjun", "Alice", "Eve"]:
>         response = ask(person, “Do you like the weather today?”, [“yes”, “no”])
> ```
> &nbsp;&nbsp;&nbsp;&nbsp;&nbsp;&nbsp; *Original instruction:* Ask ***everyone*** in the room if they like the weather today
>
> &nbsp;&nbsp;&nbsp;&nbsp;&nbsp;&nbsp; *Aligned Instruction:*  Ask ***Arjun, Alice, and Eve*** if they like the weather today
>
> 2. ***Consistency with Program Actions:*** Since the program is generated stochastically, it may modify actions that are misaligned with the original instruction. Aligned instructions can correct these discrepancies, as seen in this example:
> ```
> def task_program():
>     if not is_in_room("pen"):
>         pick("pen")
> ```
> &nbsp;&nbsp;&nbsp;&nbsp;&nbsp;&nbsp; *Original instruction:* ***Ask*** if there is a pen here. If so, pick it up.
>
> &nbsp;&nbsp;&nbsp;&nbsp;&nbsp;&nbsp; *Aligned Instruction:* ***Check*** if there is a pen here. If so, pick it up.
>
> Aligning the instruction with the program may reduce ambiguity during fine-tuning, leading to improved performance by minimizing noise and enhancing the clarity of the tasks being performed.
>
> ***Question:*** One claim made early on in the paper is that Self-Instruct gives high-diversity data but lacks correctness, and a simulator can give correctness but lacks diversity. The paper is trying to get the best of both worlds. While the current experiment results show that Robo-Instruct does generate data with high correctness, we don't see evidence that Robo-Instruct preserves diversity. Can you show some evidence (through some metrics) on diversity?
>
> ***Response:***
>
> We analyze the generated dataset in Appendix A.3. Our findings indicate that Robo-Instruct is capable of producing data with a distribution similar to that of Self-Instruct. Therefore, Robo-Instruct maintains the diversity of Self-Instruct.

---

> > ### Comment · Reviewer_afBU · 2024-08-10
> > **Response to Rebuttal**
> >
> > Thanks for the detailed response, especially clarifying the central contribution of "automatically populate simulation environments for use in verification" via Angelic Execution. A follow-up question to your example in the general rebuttal - "RoboSim will infer that apple is an object from its use with pick_up in the first line and recognize that the go_to function, which requires a location type, is inappropriately called on an object in the second line." How does your method differentiate between that (1) apple is an object that affords pick_up and go_to(apple) is erroneous and (2) apple is a location that affords go_to and pick_up(apple) is erroneous?
> >
> > Overall, the rebuttal improves my assessment of the paper contribution. However, I am still a bit skeptical whether the proposed method can scale to complex tasks that involve 20+ step executions. Additionally, best pass@1 score was around 64% for distilling GPT-4, not significantly lower than 68% (Robo-Instruct). Despite the arguments that GPT-4 is not cost-free or open-weight, I still see it as a simple and effective way of solving the same problem. Hence I am raising my score but not by a whole lot.

---

> > > ### Author Response · Authors · 2024-08-12
> > >
> > > Thank you for the improved score. We would like to offer more detailed responses to the questions raised in the comments.
> > >
> > > **Question:** RoboSim will infer that apple is an object from its use with pick_up in the first line and recognize that the go_to function, which requires a location type, is inappropriately called on an object in the second line." How does your method differentiate between that (1) apple is an object that affords pick_up and go_to(apple) is erroneous and (2) apple is a location that affords go_to and pick_up(apple) is erroneous?
> > >
> > > **Anwer:**
> > >
> > > RoboSim, infers types in the order that they are referenced. The key property it checks for is consistency --- irrespective of the ordering. Hence, if  `go_to("apple")` is called before  `pick_up("apple")`, it would guess that `apple` is of type `location`, which would be inconsistent with the subsequent call `pick_up("apple")`.
> > >
> > > If there is only a single-type use of an object, such as if only `go_to("apple")` is called, RoboSim would guess that `apple` is of type `location`, and it would not throw any errors.
> > >
> > > As in Angelic Execution[1],  RoboSim maintains an ***optimistic*** assumption of types being valid, unless they are identified to be inconsistent. In addition, RoboSim checks for violations of domain-specific constraints (e.g., trying to pick up a second object while already holding a first).
> > >
> > > **Question:** However, I am still a bit skeptical whether the proposed method can scale to complex tasks that involve 20+ step executions.
> > >
> > > **Anwer:**
> > >
> > > Our proposed method, RoboSim, automatically synthesizes simulation environments for program verification. Conceptually, it supports complex tasks with arbitrary execution steps. As the program is executed during verification, additional concepts, such as entities, types, and states, are introduced. In fact, our method benefits from tasks with more execution steps, as this provides more information about the generated program, increasing the likelihood of identifying and rejecting failing programs.
> > >
> > > Additionally, ***while solving long-horizon tasks was not an originally claimed contribution of this paper, we were intrigued by the reviewer’s inquiry and conducted a small qualitative experiment to evaluate how well the base model, Self-Instruct, Robo-Instruct fine-tuned models, and GPT-4 perform on long-horizon tasks.*** We create two instructions:
> > > 1. Let's play a game: Double and give it to the next person. Start with 1 dollar. Go to rooms A, B, C, D, E, F, and G. If you see someone, tell them how much money you have. Then ask if they would like to take the money now or double the amount and give it to the next person. If they choose to take it, the game is over, and you should come back to me. Otherwise, double your money and continue. If, in the end, no one takes the money, tell me how much you still have.
> > > 2. Go to my office and check if I have a table, a chair, and a monitor there. If any of these items are missing, go to Jason's office and see if he is there. If he is, ask him if I can borrow the missing items. If he agrees, pick up each missing item and bring it to my office. If Jason is not in his office or he says no, come back and tell me the reason.
> > >
> > > We generated the program using each model with a temperature setting of 0 and found that ***it is possible for our Robo-Instruct fine-tuned model to produce correct programs for these long-horizon tasks***, while both the base model and the Self-Instruct fine-tuned model fail. Additionally, GPT-4 made an error on the second instruction. Due to space limitations, we will share the program generated by our Robo-Instruct fine-tuned model along with GPT-4's results in a separate comment below.
> > >
> > > ***Question:*** Despite the arguments that GPT-4 is not cost-free or open-weight, I still see it as a simple and effective way of solving the same problem.
> > >
> > > ***Anwer:***
> > >
> > > We acknowledge that GPT-4 can be a viable option for generating datasets for fine-tuning domain-specific applications. However, we also emphasize that in many scenarios, ***this option is not feasible due to privacy concerns***, particularly in sectors such as financial institutions [2] and even in tech industries [3][4], despite the growing interest in using these technologies [5]. Our work offers a potential solution for these domains to fine-tune their internal models while addressing these privacy concerns.
> > >
> > > Citations:
> > > 1. Broy & Wirsing, On the Algebraic Specification of Nondeterministic Programming Languages, 1981
> > > 2. Landymore, Frank. “Giant Bank JP Morgan Bans Chatgpt Use among Employees.” Futurism, 2023
> > > 3. Gurman, Mark. “Samsung Bans Chatgpt, Google Bard, Other Generative AI Use by Staff after Leak.” Bloomberg, 2023
> > > 4. Richard, Isaiah. “Elon Musk: IOS 18 a ‘security Violation’ for CHATGPT AI Integration” Tech Times, 2024
> > > 5. Harvel N, et al., “Can LLMs Answer Investment Banking Questions?” AAAI 2024 Spring Symposium Series, 2024

---

> > > ### Author Response · Authors · 2024-08-12
> > > **Follow-up Generated Programs**
> > >
> > > ***Instruction 1***
> > >
> > > Ours:
> > > ```python
> > > def task_program():
> > >     start_loc = get_current_location()
> > >     list_of_rooms = ["A", "B", "C", "D", "E", "F", "G"]
> > >     money = 1
> > >     for room in list_of_rooms:
> > >         go_to(room)
> > >         if is_in_room("person"):
> > >             say("I have $" + str(money))
> > >             response = ask("", "Would you like to take the money now or double the amount and give it to the next person?", ["Take now", "Double and give"])
> > >             if response == "Take now":
> > >                 go_to(start_loc)
> > >                 say("The game is over, I gave the money to " + room)
> > >                 return
> > >             else:
> > >                 money *= 2
> > >     go_to(start_loc)
> > >     say("No one took the money, I still have $" + str(money))
> > > ```
> > >
> > > GPT-4
> > > ```python
> > > def task_program():
> > >     start_loc = get_current_location()
> > >     rooms = ["A", "B", "C", "D", "E", "F", "G"]
> > >     money = 1
> > >     for room in rooms:
> > >         go_to(room)
> > >         if is_in_room("person"):
> > >             say("I have " + str(money) + " dollars")
> > >             response = ask("", "Would you like to take the money now or double the amount and give it to the next person?", ["Take it", "Double it"])
> > >             if response == "Take it":
> > >                 say("Game over. The person in room " + room + " took the money.")
> > >                 go_to(start_loc)
> > >                 return
> > >             else:
> > >                 money *= 2
> > >     go_to(start_loc)
> > >     say("No one took the money. I still have " + str(money) + " dollars.")
> > > ```
> > >
> > > ***Instruction 2***
> > >
> > > Ours
> > > ```python
> > > def task_program():
> > >     start_loc = get_current_location()
> > >     go_to("my office")
> > >     table_found = is_in_room("table")
> > >     chair_found = is_in_room("chair")
> > >     monitor_found = is_in_room("monitor")
> > >     if not table_found:
> > >         go_to("Jason's office")
> > >         if is_in_room("Jason"):
> > >             response = ask("Jason", "Can I borrow the table from you?", ["Yes", "No"])
> > >             if response == "Yes":
> > >                 go_to("Jason's office")
> > >                 pick("table")
> > >                 go_to("my office")
> > >                 place("table")
> > >             else:
> > >                 go_to(start_loc)
> > >                 say("Jason said no")
> > >         else:
> > >             go_to(start_loc)
> > >             say("Jason is not in his office")
> > >     if not chair_found:
> > >         go_to("Jason's office")
> > >         if is_in_room("Jason"):
> > >             response = ask("Jason", "Can I borrow the chair from you?", ["Yes", "No"])
> > >             if response == "Yes":
> > >                 go_to("Jason's office")
> > >                 pick("chair")
> > >                 go_to("my office")
> > >                 place("chair")
> > >             else:
> > >                 go_to(start_loc)
> > >                 say("Jason said no")
> > >         else:
> > >             go_to(start_loc)
> > >             say("Jason is not in his office")
> > >     if not monitor_found:
> > >         go_to("Jason's office")
> > >         if is_in_room("Jason"):
> > >             response = ask("Jason", "Can I borrow the monitor from you?", ["Yes", "No"])
> > >             if response == "Yes":
> > >                 go_to("Jason's office")
> > >                 pick("monitor")
> > >                 go_to("my office")
> > >                 place("monitor")
> > >             else:
> > >                 go_to(start_loc)
> > >                 say("Jason said no")
> > >         else:
> > >             go_to(start_loc)
> > >             say("Jason is not in his office")
> > >     go_to(start_loc)
> > >     say("task is completed")
> > > ```
> > >
> > > GPT-4 (***incorrect***)
> > > ```python
> > > def task_program():
> > >     start_loc = get_current_location()
> > >     go_to("my office")
> > >     missing_items = []
> > >     for item in ["table", "chair", "monitor"]:
> > >         if not is_in_room(item):
> > >             missing_items.append(item)
> > >     if len(missing_items) > 0:
> > >         go_to("Jason's office")
> > >         if is_in_room("Jason"):
> > >             for item in missing_items:
> > >                 response = ask("Jason", "Can I borrow your " + item + "?", ["Yes", "No"])
> > >                 if response == "Yes":
> > >                     pick(item)              <= BUG: did not return to Jason's office after going back to my office
> > >                     go_to("my office")      <= BUG: did not return to Jason's office after going back to my office
> > >                     place(item)             <= BUG: did not return to Jason's office after going back to my office
> > >                 else:
> > >                     go_to(start_loc)
> > >                     say("Jason did not allow me to borrow his " + item)
> > >                     return
> > >         else:
> > >             go_to(start_loc)
> > >             say("Jason is not in his office")
> > >             return
> > >     go_to(start_loc)
> > >     say("Task is completed")
> > > ```

---

> > > > ### Comment · Reviewer_afBU · 2024-08-13
> > > > **Further response to rebuttal**
> > > >
> > > > I sincerely appreciate the additional comments to help provide context, clarification and present authors' arguments. I also read all other reviewers' comments. However, my opinion differs in two aspects.
> > > >
> > > > First, while I agree service mobile robots is an important/tractable area to work on/leverage recent advances in LLM/VLM, I disagree the idea of this work can be **broadly applied to other robotic applications**. In most of the mobile robot examples using LLM I have seen, actions are predefined high-level primitives (pick, place, go-to, find, rotate, say, ask). It is unlikely code-generation as described in this work will transfer well to low-level skills in embodied domains (SE(3) pose/ 6DoF grasps predictions, embodied collision avoidance, spatial reasoning, etc).
> > > >
> > > > That being said, I do believe in the value of sufficiently improving service mobile robot applications using embodied code synthesis, which I also acknowledge as indeed a non-trivial challenge. Given that the action space is quite high-level, I had thought the careful design of this method will significantly improve upon distilling GPT-4 into smaller models (in which case this paper would be an easy strong-accept). However, the only 4% improvement at best pass@1 score from 64% for distilling GPT-4 to 68% (Robo-Instruct) does not seem to be a significant advancement over simple heuristic baselines despite the clever code generation proposed in this work. There is a difference between tackling an important problem and tackling an important problem **effectively**. While VLM/GPTs should not and will not replace dedicated procedures to generate and verify code, they pose as some minimal bars for any new methods to cross to be deemed as **effective** so that the added benefits of privacy preservation, energy efficiency and fast inference can shine. Unfortunately, the results presented so far do not cross that bar for me, which is the second reason why I will hold my scores.

---

### Official Review · Reviewer_QEDf · 2024-07-13

**Soundness:** 3
**Presentation:** 4
**Contribution:** 3
**Rating:** 7
**Confidence:** 4

**Summary:**

This paper introduces a framework for generating paired instruction and robot code for further fine-tuning LLMs for robot-specific tasks. A symbolic simulator is used to check the correctness of the generated code and an LLM is prompted with chain-of-thought reasoning to align generated instruction. The resulted dataset was used to fine-tune a robot specific LLM and later tested on benchmark tasks.

**Strengths:**

This paper tackles two challenges of automated data generation for robot code synthesis, one is to check the correctness of the code by grounding it to logical state of objects and the other is to align the generated the instructions to provided robot capabilities. By designing principled and general modules that tackle each of these problems, RoboInstruct is shown to generate useful data for finetuning general purpose LLM to robot specific code generation applications.

**Weaknesses:**

RoboSIM can only check for semantically meaningful steps of the code and may not catch lower-level error that requires spatial/geometric reasoning, or even reasoning about physics, including commands that take in numerical parameters e.g. move(0,0.2,0), rotate(0.75). This seem to limit the usefulness of RoboInstruct to certain types of robot APIs.

**Questions:**

Can the prompt of InstAlign be adapted to the instruction generation step? What information after the fact is being used by instAlign that cannot be useful at initial generation round?

**Limitations:**

see weakness

---

> ### Author Rebuttal · Authors · 2024-08-06
>
> Thank you for your feedback. We would like to offer further clarification beyond the general response above to address the specific concerns and points raised in the review.
>
> ***Weakness:*** RoboSIM can only check for semantically meaningful steps of the code and may not catch lower-level error that requires spatial/geometric reasoning, or even reasoning about physics, including commands that take in numerical parameters e.g. move(0,0.2,0), rotate(0.75). This seem to limit the usefulness of RoboInstruct to certain types of robot APIs.
>
> ***Response:***
>
> RoboSim is designed to validate the correctness of programs given a particular domain --- defined by its API, types, and properties to check. We focus on task-level correctness as the domain of application, as it is of great broad interest as mentioned in the general response. However, as described in the general response, it is extensible to other domains, and we illustrate here how the domain can be easily expanded to check for lower-level actions if that is the desired level of verification.
>
> For example, consider a tabletop manipulation setting. A possible API function is `rotate(robot_gripper, radians)`.
> From the statement `rotate(“left_hand”, pi/2)`, RoboSim will infer that `left_hand` is an `entity` with the type of the robot gripper, and the state of `left_hand` is its current rotation. If there is a domain-specific constraint on the rotation of the robot gripper, such as the gripper can only rotate between $-\pi/6$ to $\pi/6$ radians, then this statement `rotate(“left_hand”, pi/2)` becomes invalid because no matter where the state rotation position of the robot gripper is, rotating $\pi/2$ radians will exceed the maximal allowable rotation range of the robot gripper: $\pi/2 > \pi/6 + \pi/6$.
>
> ***Question:*** Can the prompt of InstAlign be adapted to the instruction generation step? What information after the fact is being used by instAlign that cannot be useful at initial generation round?
>
> ***Response:***
>
> InstAlign can be applied in the initial generation round, which can occur after the Self-Instruct phase and before sending the program to RoboSim. However, as shown in Table 2, where the results compare "+Reject Unsolvable (RU)" vs. "+INSTALIGN + RU", our experiment indicates that this adaptation is not effective.
>
> The key insight here is that InstAlign proposed in Robo-Instruct will align the instruction with the validated programs, whereas adapting it at the initial generation round, the generated program could be invalid, thus limiting its effectiveness.

---

> > ### Comment · Area_Chair_nLBW · 2024-08-11
> > **Dear reviewer, please read and respond to authors' rebuttal.**
> >
> > This paper has very diverse reviews and it would benefit a lot to start a discussion to clarify any confusing points.
> >
> > Thanks!
> >
> > Your AC.

---

> ### Comment · Reviewer_QEDf · 2024-08-12
> **keeping my rating**
>
> I have read other reviewer's feedback and would like to keep my current rating.
>
> - I believe "service mobile robots" is a broad enough scope and this work makes meaningful contribution by grounding code generation for robot-specific tasks with logic. The idea of this work can be broadly applied to other robotic applications.
> - I also find it unfair to diminish the contribution of this work if we expect future generations of VLM/GPTs will be better at code generation for robots. Embodied code synthesis faces its own challenges than training generic VLMs and this paper provides a framework for safeguarding code generated automatically.

---

> > ### Author Response · Authors · 2024-08-12
> >
> > Thank you and we appreciate your recognition of this work.

---

### Official Review · Reviewer_A73B · 2024-07-16

**Soundness:** 3
**Presentation:** 3
**Contribution:** 2
**Rating:** 3
**Confidence:** 4

**Summary:**

This paper introduces ROBO-INSTRUCT, a novel framework designed to improve the code generation capabilities of smaller open-weight language models (LLMs) for domain-specific robotic tasks. ROBO-INSTRUCT leverages two key components:

1. ROBOSIM with DYNAMICEVAL:  A task-agnostic simulator that dynamically synthesizes a consistent world state based on the robot's actions within the program. This allows ROBOSIM to identify execution errors and validate generated programs even for diverse and complex tasks.
2. INSTALIGN: An instruction-program alignment procedure that utilizes Chain-of-Thought reasoning to refine the generated instructions. This ensures that the instructions better reflect the intent of the generated robot program, improving alignment between the two.

The paper evaluates ROBO-INSTRUCT by fine-tuning a Codellama-Python-7B model and testing its performance on ROBOEVAL, a benchmark for service mobile robots. The results demonstrate that the ROBO-INSTRUCT fine-tuned model significantly outperforms other open-weight models

**Strengths:**

Novel framework: Introduces ROBO-INSTRUCT, a unique approach to generating training data for fine-tuning smaller LLMs on domain-specific robot tasks.

Dynamic world synthesis: ROBOSIM's ability to dynamically create relevant world states allows it to validate diverse programs generated by SELF-INSTRUCT, overcoming the limitations of traditional simulators.

Instruction-program alignment: INSTALIGN effectively refines instructions to better reflect the program's intent, improving the quality of the training dataset.

Strong empirical results: Demonstrates that ROBO-INSTRUCT significantly improves the performance of small open-weight LLMs, enabling them to surpass even some proprietary LLMs.

Cost-effective and private:  Provides a potential alternative to deploying proprietary LLMs for local robot deployment, offering cost-effectiveness and privacy benefits.

**Weaknesses:**

Limited novelty: The idea of using a sim/emulator to verify the generated program has already been explored in previous works such as Chain-of-code, which is not mentioned by this work.

Limited scope: The paper focuses on a specific domain (service mobile robots), and it is unclear how well ROBO-INSTRUCT generalizes to other robot domains.

Lack of real-world evaluation:  The paper only evaluates ROBO-INSTRUCT on a synthetic benchmark. Real-world deployment and testing are required to further assess its practicality.

**Questions:**

Figure 1 is confusing as the authors put the overview of the proposed method, counter examples from previous methods, as well as the benchmark results all  in a single figure.

---

> ### Author Rebuttal · Authors · 2024-08-06
>
> Thank you for your feedback. We would like to provide further clarification beyond the general response above to address the specific concerns and points raised in the review.
>
> ***Weakness:*** Limited novelty: The idea of using a sim/emulator to verify the generated program has already been explored in previous works such as Chain-of-code, which is not mentioned in this work.
>
> ***Response:***
>
> The general response explains the novelty of our approach, and we focus here on the relation between RoboInstruct and Chain-of-Code (CoC), which we will cite in the paper revision.
> CoC focuses on enhancing LLM's reasoning capabilities. It *selectively simulates the interpreter* by generating the expected output of certain lines of code, which the default interpreter could not execute. The simulation procedure involves an LLM to determine valid output values that will be saved into a variable in the program. For example, given a line of the program
> ```
> answer += is_sarcastic(‘you don’t say’)
> ```
> The LLM simulator is responsible for inferring the output of the function `is_sarcastic(string)`, and determines `you don’t say` is sarcastic and subsequently returns the value of 1.
>
> However, such methods are unsuitable for the problem this work addresses. For example, consider the API function pick_up, mentioned in the general response. This API function does not return a value; instead, it updates the state of the robot in the simulation environment to indicate that the robot is holding the object. Consequently, using an LLM as the interpreter cannot simulate the state of entities in the simulation environment.
>
>
> ***Weakness:*** Limited scope: The paper focuses on a specific domain (service mobile robots), and it is unclear how well ROBO-INSTRUCT generalizes to other robot domains.
>
> ***Response:***
>
> We would like to highlight that our focus on general-purpose service mobile robots is a widely studied and popular area in the AI community as mentioned in the general response. In addition to the example of how RoboInstruct can be applied to other domains, we provide an additional example here of applying RoboInstruct to different types of robot tasks beyond service mobile robots.
> For example, consider a tabletop manipulation setting. A possible API function is `rotate(robot_gripper, radians)`.
> From the statement `rotate(“left_hand”, pi/2)`, RoboSim will infer that `left_hand` is an `entity` with the type of the robot gripper, and the state of `left_hand` is its current rotation. If there is a domain-specific constraint on the rotation of the robot gripper, such as the gripper can only rotate between $-\pi/6$ to $\pi/6$ radians, then this statement `rotate(“left_hand”, pi/2)` becomes invalid because no matter where the state rotation position of the robot gripper is, rotating $\pi/2$ radians will exceed the maximal allowable rotation range of the robot gripper: $\pi/2 > \pi/6 + \pi/6$.
>
> ***Weakness:*** Lack of real-world evaluation: The paper only evaluates ROBO-INSTRUCT on a synthetic benchmark. Real-world deployment and testing are required to further assess its practicality.
>
> ***Response:***
>
> We have deployed the Robo-Instruct fine-tuned model on a real robot using edge computing to demonstrate its real-world practicality, as illustrated in the PDF in the general response. This information will be included in the appendix of our revised paper.
>
> ***Question:*** Figure 1 is confusing as the authors put the overview of the proposed method, counter examples from previous methods, as well as the benchmark results all in a single figure.
>
> ***Response:***
>
> Thank you for the feedback. Our intention was to provide a quick summary to the reader to highlight the relation to previous approaches, our solution, and the high-level results. This can be presented with sub-figures to separate them out to make them clearer - we can make this change in the revision.

---

> > ### Comment · Area_Chair_nLBW · 2024-08-11
> > **Dear reviewer, please read and respond to authors' rebuttal.**
> >
> > This paper has very diverse reviews and it would benefit a lot to start a discussion to clarify any confusing points.
> >
> > Thanks!
> >
> > Your AC.

---

### Author Rebuttal · Authors · 2024-08-06

# General Response
We appreciate the reviewers' careful consideration and positive feedback, as well as their constructive concerns. In this response, we address common points raised in the reviews, particularly the key contribution of RoboInstruct, as well as its applicability to other domains. The PDF includes a quantitative evaluation of our model's inference speed and a real robot demonstration.

## Key contributions of this paper:

While previous studies have explored the use of simulators with ***pre-defined simulation environments*** for checking the correctness of LLM-generated output, as presented in Sec 2.1, the key contribution of this paper lies in a fundamentally new approach to ***automatically populate simulation environments*** for use in verification, which we summarize here.

A simulation environment (represented as world states in the paper) relies on three concepts:
- A list of ***entities*** to reason about, e.g., "apple", "kitchen"
- The ***type*** of the entities, and hence their affordances, e.g., "apple" is an object, you can pick it up; "kitchen" is a location, you can go to it, and it contains objects.
- The ***state*** of the entities in the world, e.g., the "apple" is in the "kitchen".

RoboSim draws inspiration from Angelic Execution [1] in software engineering previously used to infer program properties given incomplete API specifications. It automatically populates a simulation environment for each program and checks the program for correctness against this inferred environment, as presented in Alg. 3.

- First, RoboSim infers that it needs to reason about ***new entities*** when they appear in the program being checked. For instance, if a program includes the statement `pick_up("apple")`, RoboSim infers that `apple` is an entity to consider, even if it did not previously exist in the environment.

- Second, RoboSim deduces the ***type*** of an entity based on the API call used to interact with it. In the above example, `apple` is an `object`, because the `pick_up` function is called -- you can only call `pick_up` on `object` types. The domain definition outlines API functions along with their type requirements; for example, `pick_up` requires an `object` type, while `go_to` requires a `location` type. This allows RoboSim to detect inconsistencies in program interactions with entities. For example, if a program contains:
```
pick_up("apple")
go_to("apple")
```
&nbsp;&nbsp;&nbsp;&nbsp;&nbsp;&nbsp;&nbsp;&nbsp;RoboSim will infer that `apple` is an `object` from its use with `pick_up` in the first line and recognize that the `go_to` function, which requires a `location` type, is inappropriately called on an `object` in the second line. Thus, RoboSim would determine that the program fails due to this type mismatch.

- Finally, the ***state*** of the entities in the world can also affect the correct execution of the program. An example is provided in Fig 2, and here we illustrate another simple case:
```
if not is_in_room("apple"):
    pick_up("apple")
```
It is obvious to humans that the program's logic is flawed because it tries to pick up the `apple` if it isn't in the room. However, ***how would the simulator know that this is the failing state?*** The solution to this problem in RoboSim is simple: it simulates ***all possible states of all entities discovered*** and checks that none of them result in erroneous execution of the program. In this example, the state of the discovered entity `apple` can either be present in the current room or not. If the state is such that the `apple` is not present, executing the statement `pick_up("apple")` will result in an error.

Such checking would require the enumeration of many states that are exponential in the number of entities discovered. Our solution to this problem is to provide a bounded compute budget to randomly sample from this exponential space, as presented in Alg. 2.

## Applying RoboInstruct to other domains
In response to reviewers A73B and QEDf’s concerns about the project’s scope, we would like to highlight that our focus on general-purpose service mobile robots is a widely studied and popular area in the AI community [2-6]. Moreover, the key concepts in RoboInstruct are applicable to other domains.

For example, consider a broader application than robotics: code generation for an AI-powered personal digital assistant. This AI assistant could handle scheduling events using an API function like `schedule_on_calendar(event, start_time, duration)`. Given the instruction: "My schedule is free tomorrow morning. Please create two 1-hour timeslots for office hours for my robotics and deep learning class." The assistant could generate a program to create these timeslots:
```
schedule_on_calendar("robotics class office hour", “9:30 am”, “1 hr”)
schedule_on_calendar("deep learning class office hour", “10:00 am”, “1 hr”)
```
In this example, the simulator needs to reason about the entities `robotics class office hour` and `deep learning class office hour`, which are categorized as `event` types. The `event` type indicates that these entities have associated timeslots. The state of these entities is defined by the time they occur: `robotics class office hour` is set for 9:30-10:30 am, and `deep learning class office hour` is set for 10:00-11:00 am.
During evaluation, the simulator can identify a time conflict between these two office hours and thus determine that the generated program is invalid.


### References:
1. Broy & Wirsing, On the Algebraic Specification of Nondeterministic Programming Languages, 1981
2. Stark et al., Dobby: A Conversational Service Robot Driven by GPT-4, 2023
3. Li et al., Fine-Grained Task Planning for Service Robots, 2024
4. Wu et al., TidyBot, 2023
5. Wang et al., LLM-based Robot Task Planning with Exceptional Handling for General Purpose Service Robots, 2024
6. Liu et al., Ok-Robot, 2024

## PDF attachment includes experiments on inference speed and deployment to real robots

---

### Decision · Program_Chairs · 2024-09-25

**Decision:**

Reject

**Comment:**

After rebuttal, reviewers still have several concerns, including limited novelty (e.g., similar heuristics has been used, program-based verifier has been proposed in many scenarios), lack of broader domains (i.e., only code-based robotics planning in simulated world) and lack of experiments in real-world robotics scenarios. While the authors have addressed part of them, AC concludes that the paper can benefit a lot from a major revision, by incorporating reviewers' comments and doing either systematic evaluation beyond robotics domain, or going deeper into robotics domain.